# Preparedness of Nursing Homes: A Typology and Analysis of Responses to the COVID-19 Crisis in a French Network

**DOI:** 10.3390/healthcare12171727

**Published:** 2024-08-30

**Authors:** Sylvain Gautier, Fabrice Mbalayen, Valentine Dutheillet de Lamothe, Biné Mariam Ndiongue, Manon Pondjikli, Gilles Berrut, Priscilla Clôt-Faybesse, Nicolas Jurado, Marie-Anne Fourrier, Didier Armaingaud, Elisabeth Delarocque-Astagneau, Loïc Josseran

**Affiliations:** 1CESP, Inserm U1018, University of Paris-Saclay, University of Versailles Saint-Quentin-en-Yvelines, 94800 Villejuif, France; 2Department of Hospital Epidemiology and Public Health, Raymond Poincaré Hospital, Assistance Publique-Hôpitaux de Paris, 92380 Garches, France; 3Gérontopôle Autonomie et Longévité des Pays de la Loire, 44000 Nantes, France; 4Fondation Clariane, 75008 Paris, France; 5Pôle Hospitalo-Universitaire de Gérontologie Clinique, CHU Nantes, 44000 Nantes, France; 6Direction Médicale, Clariane France, 75008 Paris, France; 7Direction des Ressources Humaines, Clariane Compétences, Clariane France, 75008 Paris, France

**Keywords:** nursing homes, emergency preparedness, healthcare system resilience, primary care integration, hierarchical clustering, prevention and control measures, COVID-19 crisis

## Abstract

Background: Preparing healthcare systems for emergencies is crucial to maintaining healthcare quality. Nursing homes (NHs) require tailored emergency plans. This article aims to develop a typology of French private NHs and study their early COVID-19 responses and mortality outcomes. Methods: We conducted a cross-sectional survey among NHs of a French network consisting of 290 facilities during the first wave of the COVID-19 pandemic. A Hierarchical Clustering on Principal Components (HCPC) was conducted to develop the typology of the NHs. Association tests were used to analyze the relationships between the typology, prevention and control measures, COVID-19 mortality, and the satisfaction of hospitalization requests. Results: The 290 NHs vary in size, services, and location characteristics. The HCPC identified three clusters: large urban NHs with low levels of primary care (Cluster 1), small rural NHs (Cluster 2), and medium urban NHs with high levels of primary care (Cluster 3). The COVID-19 outcomes and response measures differed by cluster, with Clusters 1 and 2 experiencing higher mortality rates. Nearly all the NHs implemented preventive measures, but the timing and extent varied. Conclusions: This typology could help in better preparing NHs for future health emergencies, allowing for targeted resource allocation and tailored adaptations. It underscores the importance of primary care territorial structuring in managing health crises.

## 1. Introduction

Preparing healthcare systems for health emergencies and other exceptional health situations is crucial for maintaining the continuity and quality of healthcare services [1]. This is even more important in the context of the increasing and diversifying crises that healthcare systems face [2]. In this context, the concept of “resilient healthcare system” has emerged [3]. To build a resilient healthcare system, this preparation must cover all health services and various sectors involved in the care of individuals and populations. This is particularly the case for structures for the elderly that accommodate vulnerable and frail populations, such as nursing homes (NHs) and equivalent structures [4]. In France, this sector comprises just over 10,000 structures and represents around 770,000 accommodation places for the elderly (Appendix A). The vulnerability of the elderly living in NHs is often compounded by a loss of autonomy, which can significantly limit mobility in emergency situations and, more broadly, resilience [5]. Moreover, these NHs generally do not have the same resources as care structures, such as hospitals, not least in terms of healthcare workers [6]. In these community living spaces, the transmission of infectious agents can be facilitated, which may justify appropriate measures, in a constant tension between implementing restrictions and maintaining the quality of life of the residents [7].

International recommendations for the preparation of healthcare systems for exceptional health situations are primarily based on a strategic framework built around the WHO’s emergency preparedness strategic framework [8,9], complemented by regional or national strategies of different countries consolidated following successive health crises. Specifically for NHs, the preparation plans depend on healthcare systems, their specificities, and the role of the elderly care sector within these systems. Given the specific threats to this sector, preparation also relies on the implementation of thematic plans such as the one on influenza pandemics proposed by the WHO [10]. In the United States, the Centers for Medicare & Medicaid Services (CMS) has made it mandatory for NHs to develop comprehensive emergency plans, which include strategies to deal with events such as pandemics [11]. These plans must be tested and updated regularly. The Centers for Disease Control and Prevention (CDC) also provides specific guidelines for infection prevention and control in these environments [12]. Similarly, in the United Kingdom, the Care Quality Commission (CQC) is the regulatory body that ensures NHs have emergency preparedness plans. These plans must demonstrate how the facilities will protect residents and staff against various crisis scenarios. The National Health Service (NHS) also provides guidelines for crisis management [13].

In France, the preparation of NHs is primarily based on the *plan bleu* (“blue plan”, in French) developed in the context of the deadly heatwave of 2003 [14]. This plan aims to address any type of crisis to ensure the safety of residents, including natural disasters, pandemics, or other exceptional events. The law requires all NHs to have a *plan bleu*, which is developed locally under the responsibility of the NH director, although national guides have been developed to facilitate the local construction of the plan. The *plan bleu* helps to identify available resources both internally and externally and to anticipate the consequences of an identified risk. It includes measures that can be implemented to enhance the NH’s own responsiveness in emergency situations. The plan is structured around five key steps: forming a project team, analyzing and prioritizing risks, assessing response capacities, organizing crisis response, and implementing a training and exercise program. Furthermore, the *plan bleu* mandates the creation of a crisis management team and detailed procedures for triggering and lifting alerts. Specific tools and protocols are pre-established to address various risks. These elements are regularly updated and tested through drills to ensure readiness. Aside from this local plan, social and elderly care facilities are seldom mentioned in the national preparation and response plan that has been in place in France since 2014 [15].

The COVID-19 pandemic highlighted the limitations of this preparation and the response strategies. In France, where elderly people (≥65 years) were the most affected with a significant excess of deaths (+18.2%) [16], the crisis has particularly impacted NHs [17]. It affected them differently depending on their size, their own organization, and their geographic location, critical at the height of the crisis [18]. The presence of protected living units, small-scale services with limited hosting capacity that generally house elderly people with Alzheimer’s disease, was also associated with lower COVID-19 mortality. Various national restriction measures, including the general lockdown of the population—and, therefore, of NH residents—proclaimed on March 17, were implemented to reduce the spread of the epidemic and protect the NH sector (Table 1). Nevertheless, given the unprecedented nature of the situation, from the early days of the epidemic, NHs adopted new prevention or control measures prior to their adoption at the national level [19]. This resilience could reflect a specific internal organization, characteristics of the NH, or circumstantial support given the locally accessible resources in the area where the NH is located. Despite an apparent decrease in the number of hospitalizations, the hospital system was heavily burdened by COVID-19 patients, making access to care more complex, particularly for the elderly and especially for those residing in NHs [20]. Conversely, as in other contexts, interactions with the primary care sector may have been crucial in the very first moments of the crisis [21]. It now appears essential to better understand the mechanisms of this “organizational resilience” [22] in order to contribute to the continuous improvement of NH preparedness to exceptional health situations. The goal of this article is to develop a typology of NHs in a French network of private NHs and then to study the link between this typology and the early adjustment to the COVID-19 health crisis and outcomes such as mortality.

## 2. Materials and Methods

We conducted a cross-sectional survey among NHs of a French network consisting of 290 facilities during the first wave of the COVID-19 pandemic in France [18]. The statistical unit used in this work is the NH. The data used in this study come from different sources, depending on whether they were collected directly from the NHs or the network, or from other data sources.

### 2.1. Data Sources, Instruments, and Collection

Data related to the NHs were collected between 14 September 2020 and 27 October 2020, using an online questionnaire addressed only to NH directors. The questionnaire covered characteristics of the facility (capacity in terms of number of beds, area, number of buildings, types of services offered to residents, description of living, care, and technical spaces, human resources), the measures implemented during the crisis (presence and date of implementation if applicable), including in terms of personnel and supply of personal protective equipment, and finally, the number of cases and deaths of residents that occurred during the studied period. All information gathered at the facility level was validated and supplemented with administrative data available at the network headquarters (exact location of the NH, i.e., postcode of the municipality, details on human resources, aggregated information on residents: average age, average level of dependency). The data collection covered the period from March to July 2020 corresponding to the first wave of the pandemic in France. The development and administration of the questionnaire followed recommendations for online surveys [24].

Additional data used in this study to characterize the territory of each NH came from various public or scientific data sources. Four indicators describing the population at the county level where the NH was located were used. The first three were produced by the National Institute of Statistics and Economic Studies (INSEE): the proportion of people aged 75 and over (in %), the proportion of people aged 75 and over living in an NH (in %), and the urban or rural nature of the county (based on the population density of the county according to Eurostat’s NUTS3 approach). The fourth indicator concerned the number of NH beds for people aged 75 and over (in number of beds per 1000 people aged 75 and over) and was produced by the Ministry of Health.

In addition, the healthcare provision around the NH was considered at the municipal level, assuming that care for NH residents is usually provided locally. The presence of a hospital emergency service (yes/no) was obtained from the Ministry of Health’s annual health facilities statistics file (SAE). The level of territorial structuring of primary care was assessed using a dedicated typology that took into account the primary care supply, its local dynamics, and its maturity [25]. This indicator allows us to assess the strength of primary care cohesion in a given territory, understanding that the primary care sector in France is not well organized, collaborations are not systematic, and the majority of practices remain private and isolated [26]. Thus, NHs could be located in territories that were either “not structured”, had the “potential for structuring”, were “in the way for structuring”, or were “already structured”. “Already structured” is for territories where primary care is organized to address territorial public health challenges, in contrast to “under- or unstructured” territories. Some territories are “in the way for structuring”, while others are “with potential for structuring” based on the available resources in the area (such as the number of professionals and multiprofessional structures).

To characterize the magnitude of the epidemic at the territory level of each NH, we used an indicator produced by the Ministry of Health based on data from the French civil registry (INSEE). This indicator grouped the French counties according to the standardized excess mortality calculated over the period of the first wave of the epidemic (all ages combined). Three levels were considered, corresponding to an average evolution of excess mortality (in % compared to 2019 over the period from March 1 to April 20): a first group that was little or not affected (+5.2% average excess mortality in this group), a second group that was moderately affected (+44.5%), and a last group that was highly affected by an excess of mortality (+110.5%) [27]. For the analyses, the variable reflecting the magnitude of the epidemic was categorized into three classes: “low”, “moderate”, and “high”.

The impact of the pandemic was assessed based on the mortality reported by the NHs responding to the survey (a binary variable reporting the presence of at least one COVID-19 death among the NH residents) and the satisfaction of requests for hospitalization for COVID-19. This categorical variable with 3 modalities distinguished NHs for which no hospitalization request was reported and NHs for which the requests were generally satisfied (40% or more of the requests) or generally unsatisfied (less than 40% of the requests). The 40% threshold was selected based on data from the literature [28].

Concerning the prevention and control measures implemented, the variables used in the descriptive analysis were all categorical variables. The dates of implementation of certain measures were used to distinguish NHs according to whether they had implemented the measure before it was recommended nationally or after. For example, regarding the stopping of visits from families to residents, which was pronounced for all NHs in France on 11 March 2020, we distinguished NHs that had suspended visits before March 11 from those that suspended them after this date (Table 1).

### 2.2. Statistical Analysis

To develop the typology of NHs, we conducted a Hierarchical Clustering on Principal Components (HCPC) that combines hierarchical classification with factor analysis [29]. We opted for HCPC due to its suitability for handling complex data structures and relationships. This approach allows latent variables to emerge through factorial analysis before performing clustering on these latent variables. This prevents making overly strong assumptions about the links between variables. Additionally, HCPC does not require predefining the number of clusters, making it ideal for exploratory analysis, and is effective with smaller datasets and outliers, ensuring reliable and representative clusters [30]. The principal components were obtained from a multiple correspondence analysis (MCA) performed on 8 active variables describing the NHs in the network: the size of the NH (number of accommodation beds), the presence of a protected living unit within the NH (i.e., a space dedicated and adapted to the specific needs of elderly people with Alzheimer’s disease), the proportion of people aged 75 and over in the population of the NH’s territory, the proportion of people aged 75 and over living in NHs on the NH’s territory, the number of NH beds per 1000 people aged 75 and over in the NH’s territory, the presence of a hospital emergency service on the NH’s territory, and the level of territorial structuring of primary care of the NH’s territory. The quantitative variables were discretized into categorical variables using their terciles.

We retained the first 4 components using the empirical “elbow” criterion, i.e., selecting all the components from the scree plot representation just before the line flattens (Appendix A). A detailed description of the correlations between the active variables and the 4 retained components is presented in the Appendix A. From these 4 components, we obtained a 4-principal-component space. The clustering procedure within the HCPC method begins with the construction of a distance matrix between the observations in this space. Once the distance matrix is established, hierarchical ascending classification is applied using Ward’s method. This method is preferred because it minimizes intra-cluster variance, resulting in more compact and homogeneous clusters. Each observation is initially considered as an individual cluster, and the closest clusters are iteratively merged. The distances between clusters are updated by minimizing the sum of squares of the differences within each newly formed cluster. The result of this procedure is a dendrogram, which illustrates the successive cluster fusions (Figure 1), grouping NHs considered similar in their characteristics and those of the territory on which they are located.

The descriptive analysis of the relationships between the obtained typology and, on the one hand, the adaptation measures implemented, and, on the other hand, the consequences of the pandemic in terms of COVID-19 mortality and satisfaction of requested hospitalizations for COVID-19, was performed using association tests (chi-squared test or Fisher’s exact test). When the missing data represented more than 5% of the total in a category, a sensitivity analysis was conducted considering the two extreme scenarios corresponding to the assignment of one or the other of the two modalities to the missing data. The results are presented in counts and proportions in each category. The significance level considered for the tests was 0.05. The analyses were performed on *R* version 4, using *factoextra* and *FactoMineR* packages.

## 3. Results

### 3.1. A Network with Diverse Characteristics

The 290 NHs in the network are distributed throughout metropolitan France (Table 2), with nearly half of these facilities (n = 138, 47.6%) located in three administrative regions: Île-de-France (21.7%), Provence-Alpes-Côte d’Azur (13.4%), and Nouvelle-Aquitaine (12.4%). Nearly two-thirds of the network’s NHs have more than 80 beds. A large majority (n = 201, 69.3%) of the facilities have a protected living unit. The average age of residents in the network’s NHs is 88.3 years.

The territories where the NHs are located also have varied characteristics. More than half of the NHs are situated in municipalities with a hospital emergency service (n = 158, 54.5%). Nearly 60% of the NHs are located in municipal areas where the territorial structuring of primary care appears moderate: primary care is “under- or unstructured” (n = 68, 23.4%) or only has “potential for structuring” (n = 103, 35.5%). A small proportion (n = 45, 15.5%) of NHs are located in areas where the magnitude of the COVID-19 outbreak was high during the first wave, with most NHs (n = 178, 61.4%) situated in areas where the magnitude was low. Out of these 290 NHs, 192 responded to the survey, yielding a participation rate of 66.2%.

### 3.2. Three Profiles of NHs within the Network

The HCPC identified three “clusters”, leading to three profiles of NHs. The intermediate results of this modeling, particularly the study of the correlations of the principal components from the MCA, are presented in the Appendix A.

Cluster 1: This cluster comprises 86 NHs (29.7%) (Table 3). These are large facilities (>100 beds in 30.2% of cases), where residents are generally more dependent than in the other network’s NHs (average GMP of 743.5). The NHs in this cluster are mostly located in urban areas with hospital emergency services, but with a low level of primary care territorial structuring. These NHs are in areas with a low number of available NH beds and a low institutionalization rate in NHs.Cluster 2: This cluster comprises 100 NHs (34.5%). These are smaller facilities: 44.0% of them report having fewer than 80 beds. These NHs are more frequently located in rural areas than the other network’s NHs. They are in areas with a lower presence of hospital emergency services and a low level of primary care territorial structuring. In the territories where these NHs are located, the number of NH beds is within the average observed across the network, as is the proportion of institutionalized seniors over 75 years old. The magnitude of the first wave of the COVID-19 outbreak was higher in the territories of NHs in Clusters 1 and 2 than in the rest of the network.Cluster 3: This cluster comprises 104 NHs (35.9%). These are medium-sized facilities, hosting less dependent residents compared to the other network’s NHs (average GMP of 722.1). The majority of NHs in this cluster are located in areas with hospital emergency services and a high level of primary care territorial structuring. These NHs are mainly in urban areas, where the proportion of seniors over 75 years old in the population is high, as is the proportion of seniors institutionalized in NHs. The number of NH beds in these areas is higher than in the rest of the network. The magnitude of the first wave of the COVID-19 outbreak was lower in the territories of the NHs in this third cluster than in other network territories.

### 3.3. Outcomes of the Outbreak: Mortality and Hospitalization Requests

The epidemic affected the 192 responding NHs in the network differently: 81 NHs (42.2%) reported at least one resident death due to COVID-19, while 111 NHs (57.8%) reported no resident deaths due to COVID-19 during the period considered (Table 4). The proportion of NHs with at least one COVID-19 death was higher in Clusters 1 and 2 than in Cluster 3 (*p* < 0.05), where a majority of NHs reported no COVID-19 deaths (n = 50, 69.4%).

Among all the NHs that responded to the survey, just under half requested hospitalizations for one or more of their residents sick with COVID-19 during the first wave of the epidemic (n = 93, 48.4%). These hospitalization requests were generally satisfied for 81.7% of these NHs. The NHs in Cluster 1 were mostly in this situation, with 52.5% having their hospitalization requests generally satisfied. Conversely, more NHs in Cluster 2 had hospitalization requests that were not generally satisfied. The NHs in Cluster 3 mostly did not make hospitalization requests. The satisfaction of hospitalization requests for COVID-19 during the first wave of the epidemic was thus significantly associated with the profile of the NH (*p* < 0.05).

### 3.4. Prevention and Control Measures

All the NHs in the network implemented measures to contain or mitigate the COVID-19 epidemic within their facilities (Table 5). These measures were taken before or after a national recommendation was issued and made mandatory. For instance, the stopping of visits was carried out proactively by 70.6% of the NHs that responded to the survey. Similarly, 92.2% of the NHs implemented the cohorting of residents to isolate those with COVID-19 from other residents. In contrast, room confinement, involving isolating residents individually, was mostly implemented after 11 March 2020, when the recommendation was issued (n = 134, 73.2%). The same applies to mass testing, both for residents and staff, which was conducted after 6 April 2020, in most cases. An audit of practices was conducted in 73.4% of the NHs.

Among the various measures implemented by the NHs, some are significantly associated with the profile of the NH. For example, the implementation of an audit of practice was widely present in the Cluster 1 NHs and to a lesser extent (63.9%) in Cluster 3 (*p* < 0.05). The mass testing of residents and staff was generally earlier in Cluster 1, where nearly 20% of the NHs initiated testing before 6 April 2020 (*p* < 0.05).

## 4. Discussion

The main objective of this study was to design a typology of NHs within a private French network based on the environmental and organizational characteristics of the facilities. We were able to identify three profiles of NHs within the network. These three clusters of NHs are distinguished by the size of the facilities, the urban or rural nature of the territory in which they are located, and the healthcare services available in their respective territory, assessed in terms of access to a hospital emergency service and the level of territorial structuring of primary care.

Several attempts to classify NHs have been reported. Most have aimed at proposing a comparative classification of “long-term care systems”, based on the general characteristics of the NHs [31,32]. These typologies relied on qualitative methodological approaches involving subjective evaluation. Thus, while these typologies are undeniably useful for international comparisons, they differ from our approach. Bergmann et al. proposed a typology using a methodology involving MCA followed by a hierarchical classification similar to the one used in our study [33]. However, this typology focused on accommodation units (a subset of NHs), thus not allowing distinctions between the NHs themselves.

Among the studies whose main objective is to classify “substitute living place” for elderly people, Park et al., in 2006 [34], proposed a quantitative classification based on the physical and organizational environmental characteristics of the NHs. However, unlike our approach, this classification includes the characteristics of the residents. Given the fluctuating nature of these characteristics, we preferred an approach that does not consider these characteristics as modeling variables. This is also the approach favored by Lestage et al. [35] in their work on developing a Quebec classification of private NHs. This modeling work identifies five groups of NHs that differ notably in their size and the services offered to residents. In our study, we also chose to include contextual variables related to the territory of the NH to construct a typology that could reflect the ecosystem in which the NH is located to consider likely interactions. For these reasons, the typology proposed in this work is distinctly different from other typologies proposed in the literature. Furthermore, to our knowledge, this is the first attempt to develop a typology in the French context.

Another objective of our study was to examine the link between the obtained typology and the impacts of the COVID-19 epidemic (mortality, hospitalization requests) as well as the adaptation implemented (prevention and control measures). The typology is significantly associated with the adoption of certain prevention or control measures for the COVID-19 epidemic. The NHs in Cluster 1, which includes NHs located in areas with many hospital emergency services and low primary care structuring, conducted more audits of practice than the other NHs in the network. They also initiated the testing of residents and staff at an early stage of the epidemic. This could be explained by the fact that these are large NHs, likely concentrating many resources, and having access to testing earlier than other NHs in the network. Additionally, caution should be exercised in interpreting the lower proportion of NHs in Cluster 3, with hospital emergency services and a high level of primary care territorial structuring, that conducted audits of practice compared to other clusters. This may be due to the fact that mobile hygiene teams, which conducted these audits, focused, in the first wave, on the largest or most exposed facilities to the epidemic (clusters 1 and 2), as has been reported in the French context [36]. Remote training experiences for NH staff have been reported in the North American context and could be adapted to the French context [37].

Regarding the link between the typology and the impact of the COVID-19 epidemic, the NHs in the first two clusters recorded more deaths than those in Cluster 3. When they requested hospitalization, the NHs in Cluster 1 generally had their requests for COVID-19 hospitalization satisfied, whereas they were generally unsatisfied for the NHs in Cluster 2. The observed difference in mortality between Cluster 3 and the other clusters could initially be explained by the fact that the magnitude of the epidemic was lower in the NHs of Cluster 3 compared to those in the other two clusters. However, when comparing the mortality between the clusters after stratifying the magnitude of the epidemic (Appendix A), the difference in mortality remains significant, with a lower mortality in the Cluster 3 NHs. This could be partly explained by the fact that these facilities house fewer dependent residents. Another explanation lies in the fact that the level of territorial structuring of primary care in the areas where these NHs are located is higher. This high level of structuring could indeed be associated with the greater participation of primary care professionals in monitoring NH residents, as observed in Germany [38]. In other contexts, the territorial structuring of primary care has enabled the implementation of organizational adaptations based on collaboration between health professionals in the territory and NHs. The COVIDapp management and follow-up application for COVID-19 cases in NHs by general practitioners developed in Spain is an example of these organizational innovations [39]. This underscores the importance for the elderly care sector of the territorial structuring of primary care so that they can ensure the continuity of care locally before the hospital sector is called upon.

Our study has some limitations. Firstly, from a methodological standpoint, one might regret the cross-sectional nature of this study, which does not allow for the assessment of long-term adaptations. However, this is not necessary to explore the immediate reactions and emergency responses of NHs during the first moment of the crisis. Additionally, there is an information bias due to missing data for some variables. We chose to conduct a sensitivity analysis to ensure that these missing data did not affect the associations identified. Secondly, the typology work only involves a network of NHs that represents barely 4% of all the NHs in France (and 7% of the private NHs). While this limits the scope of the results, it remains a classification of NHs sharing the same governance, which should aid decision-making within the network. The chosen approach, which favors exploratory methods and clustering, could also be questioned as it condenses the information. We deemed it necessary to proceed this way to result in a parsimonious number of clusters, preserving the meaning of each obtained category. Moreover, the multivariate analysis of the impact of the COVID-19 epidemic within NHs has already been studied elsewhere [18]. The present modeling work, on the contrary, has allowed for a better understanding of the adjustments of NHs in an exceptional health situation context, thus helping to better anticipate the responses of each category of NH in the future. This will also allow for the use of this typology in etiological epidemiological studies, which could, for example, focus on the transmission of respiratory disease agents within the NHs.

In line with recommendations on organizing health emergency preparedness and response plans, it is indeed essential to consider the context, which can relate to both the organizational characteristics of the NH and the surrounding healthcare environment. This reflection is all the more important within a network of NHs to direct adequate resources and proportion preparation efforts, based on the substitution logics that can be implemented in certain territories. In the post-COVID-19 pandemic context, consolidating a resilient health system appears crucial [40]. This notably involves alliances between the different sectors of the health system (hospitals, primary care, NHs, etc.). Since January 2024, in France, NHs, multiprofessional health centers, and health territorial and professional communities (HTPCs), which are a part of the territorial structuring of primary care [25], have been able to participate in the “ORSAN system” for organizing health responses. Specifically, the HTPCs must now draft health emergency preparedness plans.

All our results also advocate for strengthening the preparedness of NHs, particularly through the consolidation of the *plan bleu*. This could be improved by incorporating tailored response plans that account for the specific characteristics of the NHs based on the cluster identified. It should promote stronger collaboration and coordination with local primary care providers and establish clear protocols and agreements with local hospitals to prioritize and streamline the hospitalization process for NH residents during crises. Additionally, the *plan bleu* should emphasize regular training for staff to enhance preparedness and responsiveness. The plan could benefit from a more detailed mapping of territorial resources and needs, which could be carried out with the help of regional health agencies, representing the Ministry of Health at the regional level.

## 5. Conclusions

By identifying three distinct clusters of NHs based on their organizational characteristics and the specificities of their territories, this study has provided a better understanding of the mechanisms behind the response to the COVID-19 epidemic within NHs. This should help in better preparing these NHs for future pandemics and exceptional health situations, allowing for the prioritization of resources and the favoring of certain adaptations, considering territorial specificities and collaborations with other territorial actors, whether in hospitals or primary care. This preparation, and the possible revision of the *plan bleu*, should be based on the expertise, opinions, and experience of the NH directors. This work also highlights the contribution of the territorial structuring of primary care in managing exceptional health situations on a territorial scale. This is all the more important in the French context, where the primary care sector has historically been based on the principle of independent, private, and isolated practice, and remains relatively unorganized to address population and public health challenges at the territorial level. The development of the territorial structuring of primary care could thus help strengthen the territorial response to exceptional health situations by involving all stakeholders and fostering an alliance with the elderly care sector.

## Figures and Tables

**Figure 1 healthcare-12-01727-f001:**
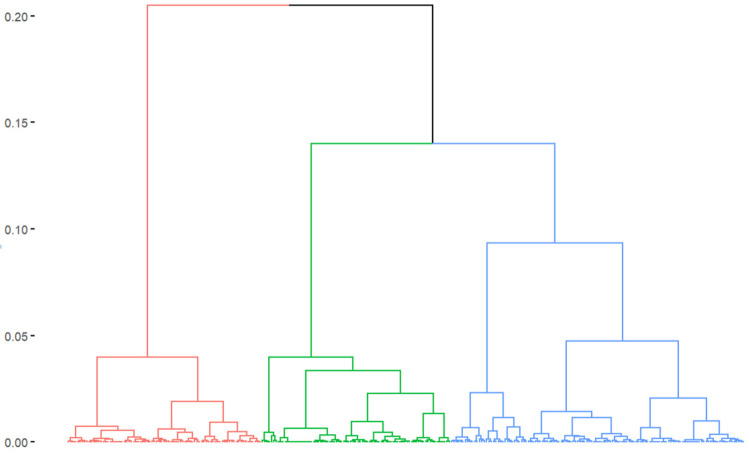
Dendrogram representing the 3 clusters of nursing homes (EHPADs) obtained through hierarchical classification. Each cluster is shown here in a different color: red for cluster 1, green for cluster 2 and blue for cluster 3. French COVID-19 nursing homes survey, 2020.

**Table 1 healthcare-12-01727-t001:** Timeline of COVID-19 pandemic management in nursing homes in France during the first wave. March to May 2020.

Date	Events/Decisions	H8 Indicator Levels from the Oxford COVID-19 Government Response Tracker [23]
Before	(None.)	0—No measures.
March 5	(None.)	1—Recommended isolation, hygiene, and visitor restriction measures in LTCFs and/or elderly people required to stay at home.
March 6	Activation of the *plan bleu* in nursing homes (national decision).	(Same as above: level 1.)
March 11	Stopping of visits in nursing homes extended to the entirety of France.	3—Extensive restrictions for isolation and hygiene in LTCFs, all non-essential external visitors prohibited, and/or all elderly people required to stay at home and not leave the home with minimal exceptions, and receive no external visitors.
March 12	Blue plan extended to all elderly care facilities (including facilities for people with disabilities).	(Same as above: level 3.)
March 17	Widespread lockdown in France.	(Same as above: level 3.)
April 1	Inclusion of deaths in nursing homes in the total count of COVID-19-related deaths.	(Same as above: level 3.)
April 1	Opinion of the national ethics advisory committee on measures concerning nursing homes and the role of professional teams (director, coordinating physician) in the implementation of lockdown.	(Same as above: level 3.)
April 6	Announcement of the initiation of screening in facilities hosting the most vulnerable individuals and professionals, primarily in nursing homes.	(Same as above: level 3.)
April 20	Reintroduction of supervised visitation rights for the elderly in nursing homes with strict adherence to barrier measures.	(Same as above: level 3.)
May 12	(None.)	2—Narrow restrictions for isolation, hygiene in LTCFs, some limitations on external visitors and/or restrictions protecting elderly people at home.

**Table 2 healthcare-12-01727-t002:** Characteristics of the nursing homes from the French network (N = 290). French COVID-19 nursing homes survey, 2020.

n (%)	N = 290
French administrative region	
Auvergne-Rhône-Alpes	30 (10.3)
Bourgogne-Franche-Comté	10 (3.4)
Bretagne	3 (1.0)
Centre-Val-de-Loire	22 (7.6)
Grand-Est	21 (7.2)
Hauts-de-France	14 (4.8)
Ile-de-France	63 (21.7)
Normandie	16 (5.5)
Nouvelle-Aquitaine	36 (12.4)
Occitanie	23 (7.9)
Pays-de-la-Loire	13 (4.5)
Provence-Alpes-Côte-d’Azur	39 (13.4)
Number of accommodation beds *	
<80 beds	109 (37.6)
80–100 beds	123 (42.4)
>100 beds	58 (20.0)
Mean age of residents (years old)	88.3
Presence of a protected living unit ^†^	201 (69.3)
Presence of a PASA ^‡^	38 (13.1)
Percentage of residents who fall ^§^	
<40%	15 (5.2)
40–50%	38 (13.1)
≥50%	237 (81.2)
Presence of a hospital emergency service in the municipality	158 (54.5)
Primary care territorial structuring (municipality level) **	
Under- or unstructured	68 (23.4)
With potential for structuring	103 (35.5)
In the way for structuring	112 (38.6)
Already structured	7 (2.4)
Number of accommodation places per 1000 people aged 75 and over in the county ^††^	
<100	74 (25.5)
100–130	150 (51.7)
≥130	66 (22.8)
Percentage of the people aged 75 and over in the county living in a nursing home	
<9.5%	198 (68.3)
≥9.5%	92 (31.7)
Percentage of people aged 75 and over in the total population of the county	
<8%	73 (25.2)
8–10%	100 (34.5)
≥10%	117 (40.3)
Magnitude of the outbreak in the county ^‡‡^	
Low	178 (61.4)
Moderate	67 (23.1)
High	45 (15.5)
Questionnaire response rate	192 (66.2)

* The average accommodation capacity of nursing homes in France is 81 beds. A proportion of 25% of nursing homes have a capacity of 100 beds or more (figures from the Ministry of Health). ^†^ Protected living units house elderly people who have been diagnosed with Alzheimer’s disease or a related disorder and who exhibit moderate behavioral problems. The care team is specifically trained to support and care for these disorders. ^‡^ A PASA (*pôle d’activités et de soins adaptés* in French) is a living space built within a nursing home to accommodate residents with Alzheimer’s disease or neurodegenerative diseases during the day. ^§^ Between 35% and 45% of people aged 65 to 90 fall every year, and this proportion rises to over 50% for those aged 90 and over (figures from the Ministry of Health). ** Typology from Gautier et al. [25]. “Already structured” is for territories where primary care is organized to address territorial public health challenges, in contrast to “under- or unstructured” territories. Additionally, some territories are “in the way for structuring”, while others are “with potential for structuring” based on the available resources in the area (such as the number of professionals and multiprofessional structures). ^††^ On average in France, there are 100 nursing home beds per 1000 people aged 75 and over (figures from the Ministry of Health). ^‡‡^ corresponds to 2020 excess of mortality compared to 2019 between 1 March and 20 April at the level of French counties [27]. “Low”, “moderate” and “high” categories correspond to counties with a mean excess of death of 5.2%, 44.5%, and 110.5% respectively.

**Table 3 healthcare-12-01727-t003:** Characteristics of the clusters of the nursing homes from the French network (N = 290). French COVID-19 nursing homes survey, 2020.

n (%)	All N = 290	Cluster 1 n = 86 (29.7)	Cluster 2 n = 100 (34.5)	Cluster 3 n = 104 (35.9)
**Active variables**				
Number of accommodation beds *				
<80 beds	109 (37.6)	24 (27.9)	44 (44.0)	41 (39.4)
80–100 beds	123 (42.4)	36 (41.9)	36 (36.0)	51 (49.0)
>100 beds	58 (20.0)	26 (30.2)	20 (20.0)	12 (11.5)
Presence of a protected living unit ^†^	201 (69.3)	57 (66.3)	68 (68.0)	76 (73.1)
Presence of a hospital emergency service in the municipality	158 (54.5)	47 (54.7)	24 (24.0)	87 (83.7)
Primary care territorial structuring (municipality level) ^‡^				
Under- or unstructured	68 (23.4)	26 (30.2)	37 (37.0)	5 (4.8)
With potential for structuring	103 (35.5)	40 (46.5)	48 (48.0)	15 (14.4)
In the way for structuring	112 (38.6)	20 (23.3)	14 (14.0)	78 (75.0)
Already structured	7 (2.4)	0 (0.0)	1 (1.0)	6 (5.8)
Number of accommodation places per 1000 people aged 75 and over in the county ^§^				
<100	74 (25.5)	61 (70.9)	5 (5.0)	8 (7.7)
100–130	150 (51.7)	25 (29.1)	85 (85.0)	40 (38.5)
≥130	66 (22.8)	0 (0.0)	10 (10.0)	56 (53.8)
Percentage of people aged 75 and over in the county living in a nursing home				
<9.5%	198 (68.3)	86 (100)	79 (79.0)	33 (31.7)
≥9.5%	92 (31.7)	0 (0.0)	21 (21.0)	71 (68.3)
Percentage of people aged 75 and over in the total population of the county				
<8%	73 (25.2)	23 (26.7)	41 (41.0)	9 (8.7)
8–10%	100 (34.5)	33 (38.4)	38 (38.0)	29 (27.9)
≥10%	117 (40.3)	30 (34.9)	21 (21.0)	66 (63.5)
Urban or rural character of the county				
Rural	23 (7.9)	2 (2.3)	19 (19.0)	2 (1.9)
Urban	267 (92.1)	84 (97.7)	81 (81.0)	102 (98.1)
**Illustrative variables**				
Mean age of residents (years old)	88.3	88.3	88.2	88.4
Mean GMP **	732.9	743.5	735.1	722.1
Percentage of wandering residents				
<20%	71 (37.0)	29 (49.2)	20 (32.8)	22 (30.6)
20–30%	69 (35.9)	18 (30.5)	21 (34.4)	30 (41.7)
≥30%	52 (27.1)	12 (20.3)	20 (32.8)	20 (27.8)
N.A.	98	27	39	32
Magnitude of the outbreak in the county ^††^				
Low	178 (61.4)	51 (59.3)	52 (52.0)	75 (72.1)
Medium	67 (23.1)	19 (22.1)	25 (25.0)	23 (22.1)
High	45 (15.5)	16 (18.6)	23 (23.0)	6 (5.8)
Questionnaire response rate	192 (66.2)	59 (68.6)	61 (61.0)	72 (69.2)

N.A.: not available. * The average accommodation capacity of nursing homes in France is 81 beds. A proportion of 25% of nursing homes have a capacity of 100 beds or more (figures from the Ministry of Health). ^†^ Protected living units house elderly people who have been diagnosed with Alzheimer’s disease or a related disorder and who exhibit moderate behavioral problems. The care team is specifically trained to support and care for these disorders. ^‡^ Typology from Gautier et al. [25]. “Already structured” is for territories where primary care is organized to address territorial public health challenges, in contrast to “under- or unstructured” territories. Additionally, some territories are “in the way for structuring”, while others are “with potential for structuring” based on the available resources in the area (such as the number of professionals and multiprofessional structures). ^§^ On average in France, there are 100 nursing home beds per 1000 people aged 75 and over (figures from the Ministry of Health). ** The GMP (for *GIR moyen pondéré* in French) reflects the average level of dependency of residents in a senior living facility. The higher this GMP, the greater the level of autonomy among the elderly. A higher GMP indicates that residents require more care, thereby justifying an increase in allocated funds. ^††^ corresponds to 2020 excess of mortality compared to 2019 between March 1st and April 20th at the level of French counties [27]. “Low”, “moderate” and “high” categories correspond to counties with a mean excess of death of 5.2%, 44.5%, and 110.5% respectively.

**Table 4 healthcare-12-01727-t004:** Mortality and satisfying hospitalization requests among the respondent nursing homes from the French network (N = 192), during the first wave of the COVID-19 outbreak, by clusters of nursing homes. French COVID-19 nursing homes survey, 2020.

n (%)	All N = 192	Cluster 1 n = 59	Cluster 2 n = 61	Cluster 3 n = 72	*p*-Value
COVID-19 mortality					<0.05
At least 1 death	81 (42.2)	28 (47.5)	31 (50.8)	22 (30.6)	
No deaths	111 (57.8)	31 (52.5)	30 (49.2)	50 (69.4)	
Satisfying hospitalization requests for COVID-19					<0.05
No requests	99 (51.5)	27 (45.8)	30 (49.2)	42 (58.3)	
Requests generally satisfied	76 (39.6)	31 (52.5)	23 (37.7)	22 (30.6)	
Requests generally unsatisfied	17 (8.9)	1 (1.7)	8 (13.1)	8 (11.1)	

**Table 5 healthcare-12-01727-t005:** Prevention and control measures implemented during the first wave of the COVID-19 outbreak in nursing homes in France, by clusters of nursing homes. French COVID-19 nursing homes survey, 2020.

n (%)	All N = 192	Cluster 1 n = 59	Cluster 2 n = 61	Cluster 3 n = 72	*p*-Value *
Stopping of visits					N.S.
Before March 11	132 (70.6)	38 (66.7)	43 (72.9)	51 (71.8)	
March 11 or later	55 (29.4)	19 (33.3)	16 (27.1)	20 (28.2)	
*Missing data*	*5*	*2*	*2*	*1*	
Room confinement					N.S.
Before March 11	49 (26.8)	12 (21.4)	13 (22.8)	24 (34.3)	
March 11 or later	134 (73.2)	44 (78.6)	44 (77.2)	46 (65.7)	
*Missing data*	*9*	*3*	*4*	*2*	
Cohorting					N.S.
Yes	177 (92.2)	54 (91.5)	57 (93.4)	66 (91.7)	
No	15 (7.8)	5 (8.5)	4 (6.6)	6 (8.3)	
Dedicated COVID-19 units					N.S.
No COVID-19 unit ^†^	73 (41.0)	17 (30.9)	24 (42.1)	32 (48.5)	
Daytime-only dedicated staff	22 (12.4)	6 (10.9)	5 (8.8)	11 (16.7)	
Nighttime-only dedicated staff	83 (46.6)	32 (58.2)	28 (49.1)	23 (34.8)	
*Missing data*	*14*	*4*	*4*	*6*	
Audit of practices					<0.05
Yes	141 (73.4)	51 (86.4)	44 (72.1)	46 (63.9)	
No	51 (26.6)	8 (13.6)	17 (27.9)	26 (36.1)	
Support by an external hygiene team					N.S.
Yes, in 2020	63 (32.8)	25 (42.4)	18 (29.5)	20 (27.8)	
Yes, but prior to 2020 or without a visit	60 (31.3)	17 (28.8)	19 (31.1)	24 (33.3)	
No	69 (35.9)	17 (28.8)	24 (39.3)	28 (38.9)	
Resident mass testing					<0.01
April 6 or before	14 (8.0)	10 (18.2)	4 (7.0)	0 (0.0)	
After April 6	161 (92.0)	45 (81.8)	53 (93.0)	63 (100)	
*Missing data* ^‡^	*17*	*4*	*4*	*9*	
Staff mass testing					<0.001
April 6 or before	11 (6.2)	9 (16.7)	1 (1.7)	1 (1.5)	
After April 6	167 (93.8)	45 (83.3)	57 (98.3)	65 (98.5)	
*Missing data* ^‡^	*14*	*5*	*3*	*6*	

* Chi-squared tests or Fisher’s exact tests. When tests are significant, it is indicated whether the *p*-value is <0.05 (chosen significance threshold), <0.01, or <0.001. Non-significant results are noted as *N.S.* ^†^ Nursing homes that did not have any COVID-19 cases among their residents did not establish dedicated COVID-19 units, as they were not affected. Nursing homes that reported opening COVID-19 units without dedicated staff were considered as not having set up dedicated COVID-19 units. Among the respondents to the survey, no nursing homes reported having established a COVID-19 unit with staff dedicated both during the day and night simultaneously. ^‡^ A sensitivity analysis was conducted considering the two extreme scenarios corresponding to the assignment of one or the other of the two modalities to the missing data.

## Data Availability

The data presented in this study are available on request from the corresponding author due to the confidential nature of some of these data.

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
