# Peer review of "Preparedness of Nursing Homes: A Typology and Analysis of Responses to the COVID-19 Crisis in a French Network"

_healthcare, 2024, doi:10.3390/healthcare12171727_

Round 1

Reviewer 1 Report (Previous Reviewer 1)

Comments and Suggestions for Authors

The authors addressed my previous points. The paper is in good shape. It can be published.

Reviewer 2 Report (Previous Reviewer 2)

Comments and Suggestions for Authors

Dear authors,

thank you for responding to my comments and editing your manuscript.

Kind regards.

Reviewer 3 Report (Previous Reviewer 3)

Comments and Suggestions for Authors

Dear Authors,

I enjoyed reading the revised version of the manuscript.

All the best!

Reviewer 4 Report (New Reviewer)

Comments and Suggestions for Authors

The COVID-19 pandemic has affected all countries and has caused serious damage to both the economy and health. The article has produced significant results for countries' health systems and crisis resilience and sheds light on health planning. There is no major issue to be corrected in the article.

This manuscript is a resubmission of an earlier submission. The following is a list of the peer review reports and author responses from that submission.

Round 1

Reviewer 1 Report

Comments and Suggestions for Authors

I read the paper with great interest. I think it contributes to the literature and it is well-written. However, there are a few issues that needs to be addressed.

1.      The paper is about the COVID-19 pandemic. However, there are not enough discussion about how the pandemic affected the elderly. For instance, the following study shows a marked reduction in healthcare utilization of the elderly.

UÄŸur, Z. B., & Durak, A. (2024). The Impact of COVID-19 on Healthcare Utilization in Turkey. Value in Health Regional Issues43, 101000.

2.      There were also recommendations that suggested targeted restrictions in response to COVID-19 pandemic. The paper did not talk about these aspects in detail. What France did during the COVID-19 pandemic is also not well-explained. I saw after finishing the paper and checking the supplemantary material. For instance, were the casualties in France due to COVID-19 mostly elderly? How France’s COVID-19 casualities compare with other countries?

and the level of territorial structuring of primary care on the NH’s territory”

This is not clear to me.

A small proportion (n = 45, 15.5%) of NHs are located in areas where the magnitude of the COVID-19 outbreak was high during the first wave”

-What does it mean that COVID-19 outbreak was high? High compared to what?

“Requests generally satisfied” Satisfying hospitalization requests for COVID-19.

I cannot understand what it means to satisfy hospitalization requests. I guess the NH request to hospitalize some of the residents. Is it possible not to send them to hospital?

Can hospitals choose not to receive these patients? How should we read these statistics about satisfaction of hospitalization requests?

Table S-1 is very useful but I could not understand the sign “–“ put for March 6 for H8 indicator. Does this mean that it is missing or is it also 1?  March 12 to May 12 is also marked as “-“.

  As your main aim is to categorize NHs, you may consider putting the screeplot in the main text. Only a suggestion.

Conclusion is not really conclusive to me. What did we learn from this research? 

Author Response

Comments 1: I read the paper with great interest. I think it contributes to the literature and it is well-written. However, there are a few issues that needs to be addressed.

Reponse 1: We thank the reviewer for his comments and suggestions and appreciate that he highlights the value of our approach.

Comments 2: 1. The paper is about the COVID-19 pandemic. However, there are not enough discussion about how the pandemic affected the elderly. For instance, the following study shows a marked reduction in healthcare utilization of the elderly.

UÄŸur, Z. B., & Durak, A. (2024). The Impact of COVID-19 on Healthcare Utilization in Turkey. Value in Health Regional Issues43, 101000.

Response 2: The reviewer is absolutely right regarding the need for precision about the mortality consequences of the first wave of the COVID-19 epidemic in France, particularly among the elderly people. To better contextualize this point, we have added a reference (A Fouillet, 2020) that details mortality by age groups in France during this period, in the introduction section.

Moreover, we have consulted the suggested reference with great interest. It indeed shows, in the Turkish context, a decrease in healthcare system utilization. This was also widely observed during the first wave of the epidemic in Europe and North America (see Rosenbaum L. The untold toll—the pandemic’s effects on patients without COVID-19. N Engl J Med. 2020 and Moynihan R et al. Impact of COVID-19 pandemic on utilisation of healthcare services: a systematic review. BMJ Open). Equivalent studies have also been conducted in France and have highlighted a decrease in hospitalizations during the first wave, particularly during the lockdown. We have chosen to include one of these studies (M J Torres, 2023) and mention this point in the introduction section. The authors emphasize that this decrease in hospitalizations for the elderly, along with an increase in mortality in this age group, could be linked to delays in accessing care and the healthcare system’s inability to provide an amount (and quality) of care equivalent to what it usually produces (saturation effect). We have thus added this to the introduction section as follows: “Despite an apparent decrease in the number of hospitalizations, the hospital system was heavily burdened by COVID-19 patients making access to care more complex, particularly for the elderly and especially for those residing in NHs (M J Torres, 2023).”

Comments 3: 2. There were also recommendations that suggested targeted restrictions in response to COVID-19 pandemic. The paper did not talk about these aspects in detail. What France did during the COVID-19 pandemic is also not well-explained. I saw after finishing the paper and checking the supplemantary material. For instance, were the casualties in France due to COVID-19 mostly elderly? How France’s COVID-19 casualities compare with other countries?

Response 3: Thank you for these suggestions. Regarding the measures implemented in nursing homes (and outside) in France, we have indeed chosen to include them in the appendices, but as you suggested, Table S1 could be added directly to the main body of the article. This table notably mentions the date of the general lockdown that was implemented to limit the spread of the epidemic. We leave it to the editor to judge the relevance of this addition. We have added a sentence to clarify this point and reference the table in the introduction section. However, we want to emphasize that the measures implemented, as mentioned in the introduction, could be anticipated by the nursing homes and implemented by some even before the measure was mandated nationally and applied to all nursing homes in France. It is precisely this anticipation and adaptation “beyond recommendations” that we explore in this article through the resilience capacity of nursing homes in an unprecedented health situation (at the beginning of the crisis, during the first wave).

Regarding COVID-19 mortality among the elderly, it was the highest (with an excess mortality of +18%). We have added a reference and a sentence to specify this in the introduction section.

Comments 4:  “and the level of territorial structuring of primary care on the NH’s territory”
This is not clear to me.

Response 4: We would like to thank you the reviewer for his comment. The “level of territorial structuring of primary care” corresponds to the degree of strengthening of the primary care sector at the territorial level. It is a composite indicator that includes the current supply, its longevity, and its dynamics over the past five years, developed elsewhere (Gautier & Josseran, 2024). We use this indicator here to reflect the existing cooperation and collaboration logics between professionals and primary care structures (healthcare centers, for instance) within a given territory, in a French healthcare system still largely dominated by private and isolated practice. This allows us to estimate the capacity of primary care in a given territory to organize and respond to the population’s needs. To clarify this and make the manuscript clearer for the reader, we have added a sentence in the methodology section: “This indicator allows us to assess the strength of primary care cohesion in a given territory, understanding that the primary care sector in France is not well-organized, collaborations are not systematic, and the majority practice remains private and isolated”. We have also added a reference that describes the organization of primary care in France. Furthermore, the text you mentioned is indeed difficult to understand. This is due to a language error. We have changed it to clarify.

Comments 5: “A small proportion (n = 45, 15.5%) of NHs are located in areas where the magnitude of the COVID-19 outbreak was high during the first wave”

-What does it mean that COVID-19 outbreak was high? High compared to what?

Response 5: We thank you the reviewer for his question. To quantify the magnitude of the epidemic at the territorial level, as specified in the methods section, we used an indicator that distinguishes three levels of excess mortality. Thus, a territory where the intensity of the epidemic was “high” corresponded to a territory with an average excess mortality of +110.5%. This assessment is based on excess mortality, measured, as specified, in comparison to previous years. To clarify this point, we have added a sentence in the methods section to specify that the magnitude of the epidemic was thus classified as “low,” “moderate,” or “high.” It should also be noted that the variable magnitude is due to the fact that the COVID-19 epidemic during the first wave did not affect France uniformly.

Comments 6: “Requests generally satisfied” Satisfying hospitalization requests for COVID-19.

I cannot understand what it means to satisfy hospitalization requests. I guess the NH request to hospitalize some of the residents. Is it possible not to send them to hospital?

Can hospitals choose not to receive these patients? How should we read these statistics about satisfaction of hospitalization requests?

Response 6: We understand the reviewer’s remark. Indeed, during the COVID-19 pandemic, emergency services, through which patients needing hospitalization are admitted to the hospital, were heavily utilized and to some extent overwhelmed by COVID-19 patients. Given the significant number of patients requiring respiratory care, often in intensive care units, it is possible that hospitalization requests from nursing homes were deemed less of a priority due to the advanced age or the clinical situation of the patients concerned (to avoid a form of unreasonable obstinacy of care). This was particularly evident at the peak of the epidemic and has been documented, for example, in the Italian context (see Orfali K. What Triage Issues Reveal: Ethics in the COVID-19 Pandemic in Italy and France. J Bioeth Inq. 2020).

Comments 7: Table S-1 is very useful but I could not understand the sign “–“ put for March 6 for H8 indicator. Does this mean that it is missing or is it also 1?  March 12 to May 12 is also marked as “-“.

Response 7: We thank you the reviewer for his remark. Indeed, the symbol “-” indicated the maintenance of measures as they were. It does not represent missing data. Thus, regarding the H8 indicator, it went directly from level 1 to level 3, without previously passing through level 2. We have modified the table accordingly to make this clearer. Additionally, we appreciate your suggestion to add this table to the manuscript. We have made this change as previously mentioned.

Comments 8:  As your main aim is to categorize NHs, you may consider putting the screeplot in the main text. Only a suggestion.

Response 8: Thank you for this suggestion. We have chosen not to include the scree plot in the manuscript to avoid overloading the content. This is all the more important now that we have included Table S1 in the manuscript (and a Figure representing the dendrogram). However, we can certainly integrate it into the manuscript. We leave it to the editor to decide if this is necessary.

Comments 9: Conclusion is not really conclusive to me. What did we learn from this research? 

Response 9: Thank you for this comment. We aimed to remain relatively cautious regarding the conclusions about the work done, mainly emphasizing the contribution of the typology in preparing for future responses. This is particularly to better consider the deployment of resources (both material and human) or prioritize certain nursing homes over others given their internal organizational situation and the configuration of their territory. Specifically, we wanted to stress the territorial dimension, echoing WHO recommendations advocating for resilient health systems at the local level. Thus, we highlighted the importance, in our view, of the territorial structuring of primary care, which we suggest could partially compensate for deficiencies in the hospital system. This compensation should be considered in the French context, where, unlike primary care, the hospital sector is administered by public authorities. The primary care sector remains relatively unorganized and under-administered, with practice predominantly isolated, following the principle of private practice. To emphasize this point, we have added a few sentences to the conclusion, as follows: “This is all the more important in the French context, where primary care sector has historically been based on the principle of independent, private and isolated practice, and remains relatively unorganized to address population and public health challenges at the territorial level. The development of territorial structuring of primary care could thus help strengthen the territorial response to exceptional health situations by involving all stakeholders and fostering an alliance with the elderly care sector.”

Reviewer 2 Report

Comments and Suggestions for Authors

I am including my comments as an attachment.

Author Response

Comments 1: The aim of this study was to develop a typology of French private nursing homes (NHs) and examine their early responses to the COVID-19 pandemic and associated mortality outcomes. The research included a cross-sectional survey of 290 facilities during the first wave of the pandemic, with data analyzed using hierarchical clustering. The study identified three distinct clusters of NHs based on organizational characteristics and territorial specifics, providing a deeper understanding of the mechanisms behind NHs' responses to the COVID-19 epidemic. The main contributions of this work are the revelation of differences in mortality outcomes and response measures among the clusters, which can aid in targeted resource allocation and adaptations in future health crises. The strengths of the study lie in its ability to offer a detailed typology and its potential application for improving NH preparedness and resilience to exceptional health situations.

Response 1: We would thank the reviewer very much for his interest in our work. We greatly appreciate his thoughtful insights and suggestions.

Comments 2: What is the main question addressed by the research?

The authors sought to understand how different organizational characteristics and territorial specifics influence the ability of NHs to respond to exceptional health situations. The study aimed to identify key factors that contribute to the resilience and effectiveness of NH responses to the pandemic and provide insights that could be used to better prepare these facilities for future health crises, allowing for targeted resource allocation and adaptation of preventive and control measures to the specific needs of different types of NHs.

Response 2: This comment from the reviewer does not require a specific response from us.

Comments 3: Commentary on the concept:

The authors present the limitations and weaknesses of the research study in the secondto-last paragraph of the results section, providing justification for their occurrence. A significant shortcoming of the manuscript is the failure to adhere to the standard IMRD (Introduction, Methods, Results, and Discussion) structure of scientific articles, as the discussion section is missing. The absence of this section limits the ability to critically evaluate the findings, interpret them, and contextualize them in relation to existing literature and practical implications.

Response 3: Could the reviewer please clarify the point regarding the absence of a discussion section? We agree with the reviewer that a discussion section is absolutely necessary in order to “evaluate the findings, interpret them, and contextualize them in relation to existing literature and practical implications”. Hence, we of course provided a discussion section presenting the main results of our work and discussed regarding other typologies identified in the literature. We remain available to the reviewer to clarify the elements presented in the discussion section.

Comments 4: Other commenting:

The topic covered in the manuscript is relevant, especially given the current global emphasis on improving healthcare systems' preparedness for pandemics and other exceptional health situations. The focus on the typology of French private nursing homes (NHs) and their responses to COVID-19 is particularly timely and significant. However, the completeness of the review topic could be improved.

Response 4: Thank you to the reviewer for noting that the topic covered in the manuscript is relevant. Could the reviewer please clarify what he means by “the completeness of the review”? As suggested, we have added contextual elements about the elderly care sector in France. We hope these additions will meet the reviewer’s expectations.

Comments 5: While the manuscript provides a good overview of the different types of NHs and their early responses to the pandemic, it lacks a comprehensive discussion on the long-term implications of these responses and how they might inform future preparedness strategies.

Response 5: Thank you for this valuable feedback. Could the reviewer please clarify what he means by “long-term” in this context? We did not directly study the impacts of the measures on the pandemic and limited our research to the period of the first wave, where we could observe the phenomenon of resilience (an adaptation to cope with the unexpected) in nursing homes.

Comments 6: A notable gap in knowledge identified in the manuscript is the limited exploration of the specific mechanisms and factors within each identified cluster that contributed to their respective outcomes. For instance, while the study highlights differences in mortality rates and preventive measures, it does not delve deeply into the underlying reasons for these differences, such as variations in staff training, resource availability, or leadership practices.

Response 6: We thank you the reviewer for his remark. We have included discussion elements on the reasons that may explain the adaptations or the impacts of the epidemic according to the clusters. These elements are included in the discussion section of the manuscript. For instance, we mentioned an American initiative for staff training, as follows: “Remote training experiences for NHs’ staff have been reported in the North American context and could be adapted to the French context (Baughman AW et al., 2021).”

Comments 7: The references used in the manuscript are generally appropriate and relevant to the topic. However, there could be a broader inclusion of recent literature on the organizational resilience of healthcare facilities and the role of primary care networks in pandemic response. Including more international studies and comparisons would also enhance the manuscript’s robustness and provide a more comprehensive understanding of the subject.

Response 7: Thank you to the reviewer for his comment. It was very difficult for us to identify literature addressing the resilience of nursing homes during the COVID crisis, particularly in relation to the primary care sector. While there are many articles on the response of primary care to the pandemic, to our knowledge, they do not address how primary care interacted with nursing homes or continued to care for their vulnerable patients. We particularly relied on the article by Kühl et al., which discusses the German context. In their discussion, the authors also note a lack of studies on interactions between primary care and nursing homes during the first wave of the epidemic in Europe. Furthermore, we would like to remind that our objective was not to study the consequences of the epidemic on nursing homes or residents, nor to study the effects of adopting certain prevention and control measures on mortality or hospitalization requests. Our aim was to determine a typology of nursing homes that could be used to prioritize future actions in the event of a health crisis.

Comments 8: In terms of scientific content, the manuscript would benefit from a more detailed methodology section that explains the rationale behind the choice of hierarchical clustering and how it was specifically applied. Additionally, the results section could be expanded to include more granular data, such as specific preventive measures taken by each cluster and their direct impact on resident outcomes.

Response 8: We thank the reviewer for these suggestions. Indeed, we opted for hierarchical clustering on principal components (HCPC) for several reasons. Firstly, HCPC is well-suited for handling complex data structures and relationships, which is the case in our study. This approach allows latent variables to emerge through factorial analysis (in this case, MCA) before performing clustering on these latent variables. This prevents making overly strong assumptions about the links between variables, especially since, in the case of nursing homes, variables are often highly associated with each other (e.g., size is a latent variable that determines many other variables such as the presence of a protected living unit). Furthermore, unlike some clustering methods that require the number of clusters to be defined in advance (e.g., k-means), HCPC does not necessitate prior knowledge of the number of clusters. This allows for a more exploratory approach to understanding the natural groupings within the data. Given our sample size of 290 NHs, HCPC is appropriate as it performs well with smaller datasets, unlike some other clustering techniques that may require larger sample sizes for stability and accuracy. Finally, HCPC is robust to variations in the data and can handle outliers effectively. This is important in our study to ensure that the identified clusters are representative and reliable. For all these reasons, we have favoured this methodological approach. As suggested by the reviewer, we have added explanatory elements in the methods section, to justify the use of a HCPC approach.

Regarding the results section, we understand the reviewer’s comment. It is likely an error in the positioning of Table 5 (formerly Table 4) in the manuscript, which provides the requested details (the measures implemented by each cluster of nursing homes). However, it is not possible for us to study the impact of each of these measures on the outcomes studied (mortality and hospitalization requests), mainly due to the cross-sectional nature of this study. Indeed, although we have the date of implementation of these measures, most nursing homes eventually implemented all (or almost all) of the cited measures. Therefore, in the absence of a control group, it is not possible to conjecture about the consequences of these measures for the residents. We have repositioned Table 5 in the manuscript.

Comments 9: Commentary on the material and methods used.

The methodological approach of the manuscript demonstrates several strengths. The use of a cross-sectional survey to gather comprehensive data on the characteristics and responses of nursing homes (NHs) during the COVID-19 pandemic is appropriate and provides valuable insights. The application of Hierarchical Clustering on Principal Components (HCPC) is also a notable strength, allowing for the identification of distinct clusters of NHs based on various organizational and territorial variables.

Response 9: Thank you very much for this comment highlighting the relevance of the methods used, particularly the HCPC, which allows for reducing the dimensionality of the data through multiple correspondence analysis (MCA) and then applying hierarchical clustering to identify natural groupings within the data. This method is especially useful in revealing patterns and relationships between nursing homes based on their organizational characteristics and territorial specifics.

Comments 10: However, there are areas for improvement. The cross-sectional design limits the ability to capture dynamic changes over time, which is crucial for understanding evolving responses to the pandemic. Additionally, the selection of variables for the HCPC could be expanded to include more relevant factors, such as staff infection rates and specific departmental measures. The manuscript could also benefit from a more detailed discussion of potential biases due to missing data and the robustness of the clustering results, potentially validated through alternative methods. Overall, while the methodological framework is solid, addressing these limitations could enhance the study's robustness and reliability.

Response 10: We agree with this remark regarding the limitations of a cross-sectional study. However, in this work, we do not aim to study the evolution of measures taken by nursing homes or their long-term consequences, but rather to identify those that were implemented during the first wave of the epidemic. Our goal is to explore the immediate reactions and “emergency” responses to the health crisis and, in the absence of systematic national recommendations, to understand the resilience of nursing homes. We aim to determine whether this resilience is linked to the type of nursing home, i.e., to organizational factors specific to the nursing home and the characteristics of its territory.

Regarding the variables used to develop the typology, we deliberately did not include variables related to the response to the COVID-19 epidemic because we wanted a typology based “a priori” on the characteristics of the nursing home and its territory “before the crisis.” Additionally, we aimed to retain only elements that do not vary substantially to ensure a certain “stability” in the classification obtained. Thus, staff numbers (which can change due to potential departures) and resident numbers (or mean age of the residents) were not included in the typology.

Regarding the missing data, we conducted a sensitivity analysis to ensure the influence of these missing values on the regressions. We chose to study the two extreme scenarios by assigning all missing values the same category to create a potential imbalance that could alter or negate the direction of the association. This was not the case. We have clarified this point in the methods section and in a footnote of the table. We appreciate the suggestion to better clarify the methodological strengths of this approach and to discuss its validity. We have incorporated this point into the discussion section as follows: “From a methodological standpoint, one might regret the cross-sectional nature of this study, which does not allow for the assessment of long-term adaptations. However, this is not necessary to explore the immediate reactions and emergency responses of NHs during the first moment of the crisis. Additionally, there is an information bias due to missing data for some variables. We chose to conduct a sensitivity analysis to ensure that these missing data did not affect the associations identified.”

Comments 11: Additional comments

Chapter 3 should contain two or three sentences on what it deals with, what it presents. Then only start chapter 3.1.

Response 11: We may not be completely clear on what the reviewer means about “two or three sentences on what the chapter deals with”. Should we introduce a sentence to detail the outline of the section? We will leave it to the editor to judge if this is the journal’s standard practice.

Comments 12: Are the references appropriate?

The literature referred to by the authors is relevant.

I recommend it for publication with minor modifications.

Response 12: We thank the reviewer for this comment.

Reviewer 3 Report

Comments and Suggestions for Authors

Dear Authors,

I thoroughly went through your manuscript which deals with an important topic relating to health protection of old adults living in nursing homes. In the following please find my comments:

Introduction:

Please add a paragraph on the provision structure of nursing homes in France (e.g. number of facilities, geographical contexts (urban-rural), size). What about public NHs? Shift relevant parts of the text from the discussion chapter to this paragraph.

Please provide more information on the NHs’ “plan bleu”.

Please specify the aim of the article. In my point of view this piece of work is “an extension” of previous work by some of the authors of this article (reference 17) with the difference that the “same” sample of 192 NHs was analysed using a hierarchical cluster analysis.

Material and Methods:

The chapter needs particular attention:

Use of inaccurate terms in order to classify territories: “not structured”, “potential for structuring” etc. (line 131ff).

Much of the information listed in tabular form in the supplements needs to be included here.

Hierarchical cluster analysis: information on procedure is missing.

Results:

The presentation needs further improvement: too many tables, formatting of text.

Discussion:

The discussion of the empirical results lacks a conceptual or theoretical context. In my view, French’s plan bleu would be a suitable.

Conclusions:

The statements are too general. Regarding future research needs, consideration should be given to discussing the results with the NHs directors.

Further comment:

Please explain the term “medico-social sector”.

All the best.

Author Response

Comments 1: I thoroughly went through your manuscript which deals with an important topic relating to health protection of old adults living in nursing homes. In the following please find my comments:

Response 1: We thank the reviewer very much for his thorough reading and constructive remarks to improve the manuscript.

Comments 2: Introduction:

Please add a paragraph on the provision structure of nursing homes in France (e.g. number of facilities, geographical contexts (urban-rural), size). What about public NHs? Shift relevant parts of the text from the discussion chapter to this paragraph.

Response 2: We thank the reviewer for his comment. We recognize the importance of providing a detailed overview of the provision structure of nursing homes in France, including the number of facilities, geographical contexts (urban-rural), and size, as well as the role of public nursing homes. However, such a synopsis may be too extensive to present in the introduction section of the manuscript. We suggest adding supplementary material to describe the organization of nursing homes in France comprehensively. We have also added a clarification regarding the scope represented by the 290 NHs studied, indicating that they represent 7% of all private NHs.

Comments 3: Please provide more information on the NHs’ “plan bleu”.

Response 3: We thank the reviewer for this suggestion. We have added these additional elements to describe the plan bleu in the introductory section: “The plan is structured around five key steps: forming a project team, analyzing and prioritizing risks, assessing response capacities, organizing crisis response, and implementing a training and exercise program. Furthermore, the plan bleu mandates the creation of a crisis management team and detailed procedures for triggering and lifting alerts. Specific tools and protocols are pre-established to address various risks. These elements are regularly updated and tested through drills to ensure readiness.”

Comments 4: Please specify the aim of the article. In my point of view this piece of work is “an extension” of previous work by some of the authors of this article (reference 17) with the difference that the “same” sample of 192 NHs was analysed using a hierarchical cluster analysis.

Response 4: We thank the reviewer for this thorough reading. We confirm that is indeed an original article, never published elsewhere. Its objective is not related to the authors’ previously conducted work. It is a unique modeling effort aimed at developing a typology of nursing homes based on data from a sample of 290 NHs (from a private network), whereas previous work only involved a smaller sample of 192 NHs that responded to the mentioned survey. As you noted, we do reuse data from the previously conducted survey, but the prior work aimed to study the consequences of the COVID-19 crisis on NHs. Here, the objective is to understand how NHs, for which we determine a typology a priori, adapted in the very early stages of the epidemic. In other words, it is about studying their resilience in the face of an unexpected crisis. We cited the previous article right from the introduction to highlight what this first study revealed and took care to clearly distinguish in the discussion section the objectives pursued here in this work from the conclusions of the previous data exploitation: “the multivariate analysis of the impact of the COVID-19 epidemic within NHs has al-ready been studied elsewhere [17]. The present modeling work, on the contrary, has allowed for a better understanding of the adjustments of NHs in an exceptional health situation context, thus helping to better anticipate the responses of each category of NH in the future.”.

Comments 5: Material and Methods:

The chapter needs particular attention:

Use of inaccurate terms in order to classify territories: “not structured”, “potential for structuring” etc. (line 131ff).

Response 5: We want to thank the reviewer for his comment. Indeed, we are aware that the terms used may not be sufficiently clear for the reader. These terms directly refer to the typology used, derived from previously published work (as mentioned), which distinguishes territories based on the degree of primary care structuring, in other words, the level of collaboration and cohesion that exists between the actors and primary care structures in the territory. We have added several sentences to better introduce and contextualize this indicator in the method section. It is important to recall that primary care in France remains relatively unorganized and under-administered, and their organization is ultimately left to the discretion of the professionals themselves, who still primarily practice privately and generally alone. The logics of grouping and coordinated practice that have been in place for a few years lead to territorial differentiations: some territories appear very clearly “structured” in terms of primary care, meaning they are organized to address public health issues present in their territory, while others are not yet, and some are in intermediate phases “on the way to being” or “with the potential to be,” particularly because they have promising resources in the area (a high number of professionals, newly created multiprofessional structures, etc.). We hope that these clarifications will make the manuscript clearer. We have also added a sentence under each table to precise it: “Already structured” is for territories where primary care is organized to address territorial public health challenges, in contrast to “under or unstructured” territories. Additionally, some territories are “in the way for structuring,” while others are “with potential for structuring” based on the available resources in the area (such as the number of professionals and multiprofessional structures).

Comments 6: Much of the information listed in tabular form in the supplements needs to be included here.

Response 6: Thanks to the reviewer for the suggestion. However, could it be possible to specify the nature of the supplementary material information the reviewer is referring to and that should be included in the text? On the advice of another reviewer, we have integrated Table S1, which presents the measures implemented at the national level and their history during the first wave. We hope this will be satisfactory.

Comments 7: Hierarchical cluster analysis: information on procedure is missing.

Response 7: Thanks to the reviewer for this remark. To better clarify the method employed, we have both explained the choices that justify the use of the HCPC method and detailed, as requested, the section on clustering, as follows: “After obtaining the principal components via Multiple Correspondence Analysis (MCA), the clustering procedure within the HCPC method begins with the construction of a distance matrix between the observations in the principal component space. Once the distance matrix is established, hierarchical ascending classification (HAC) is applied using Ward's method. This method is preferred because it minimizes intra-cluster variance, resulting in more compact and homogeneous clusters. Each observation is initially considered as an individual cluster, and the closest clusters are iteratively merged. The distances between clusters are updated by minimizing the sum of squares of the differences within each newly formed cluster. The result of this procedure is a dendrogram, which illustrates the successive cluster fusions (Figure 1).” We added the dendrogram as a Figure in the manuscript.

Comments 8: Results:

The presentation needs further improvement: too many tables, formatting of text.

Response 8: Thanks to the reviewer for this comment regarding the results section. We believe that all the presented tables are essential and support the various stages of the results. However, we are aware that their format could be improved, as proposed in the modified version submitted. We leave it to the editor to indicate which tables, if any, should be moved to the appendix.

Comments 9: Discussion:

The discussion of the empirical results lacks a conceptual or theoretical context. In my view, French’s plan bleu would be a suitable.

Response 9: We agree with the reviewer’s remark. As requested earlier, we have further introduced the plan bleu and, following their advice, we have included a discussion point in the discussion section. This is particularly to emphasize the importance of these results in proposing improvements to the plan bleu. The sentences added are: “All our results also advocate for strengthening the preparedness of NHs, particularly through the consolidation of the plan bleu. It could be improved by incorporating tailored response plans that account for the specific characteristics of the NHs based on the cluster identified. It should promote stronger collaboration and coordination with local primary care providers and establish clear protocols and agreements with local hospitals to prioritize and streamline the hospitalization process for NH residents during crises. Additionally, the plan bleu should emphasize regular training for staff to enhance preparedness and responsiveness. The plan could benefit from a more de-tailed mapping of territorial resources and needs, which could be carried out with the help of regional health agencies, representing the Ministry of Health at the regional level.”

Comments 10: Conclusions:

The statements are too general. Regarding future research needs, consideration should be given to discussing the results with the NHs directors.

Response 10: We would like to thank the reviewer for his comment. Indeed, the suggestion is very interesting and accurate. We have added the following statement to the conclusion: “This preparation, and the possible revision of the Plan Bleu, should be based on the expertise, opinions, and experience of the NH directors.” Furthermore, we have strengthened the conclusion by emphasizing the relevance of our results in the context of the relatively unorganized and under-administered French primary care system.

Comments 11: Further comment:

Please explain the term “medico-social sector”.

Response 11: The term “medico-social sector” is an incorrect translation of the French term “secteur médico-social,” which refers to the entirety of services and facilities that provide support and care for individuals in vulnerable situations, whether due to age, health status, disability, or social circumstances. This sector includes nursing homes. To be more precise and to emphasize this segment of elderly housing, we have chosen to change this expression to “elderly care sector”.